# Modification of Disinfection Process at a Local Water Treatment Plant—Skawina (Poland)

**Bogumiła Winid** [1,*] , **Robert Muszański** [2] **and Jan Wilkosz** [3,†]

1 Faculty of Drilling, Oil and Gas, AGH University of Science and Technology, Mickiewicza 30 Ave., 30-059 Krakow, Poland
2 Wofil Ozone Technology, Rzeźniana 10/1 St., 33-380 Krynica-Zdrój, Poland
3 Independent Researcher, 34-114 Brzeźnica, Poland
* Correspondence: winid@agh.edu.pl; Tel.: +48-126-1725
† Retired.

**Abstract:** This paper summarizes studies undertaken at a water treatment plant in Skawina (WTP Skawina) where the disinfection process was modified by introducing a mobile ozonation system. The application of a small-size, fully-automated ozonation installation only slightly complicates the water treatment process, without the need to redesign the water treatment line, and with relatively low investment costs. The aim of this study was to analyze whether the change of the disinfection method affects the final water quality. The investigated water samples were treated in the mobile ozonation system using a disinfection process with only sodium hypochlorite. Treated water was of excellent quality, and seasonal variations in raw water parameters (variable organic matter contents) did not result in elevated trihalomethanes (THM) and bromate concentrations. Despite the trace amounts of bromides in the water prior to treatment, the water in the municipal drinking water system did not contain determinable amounts of bromates. The bromine concentrations in the treated water supplied to the water distribution system were higher than in raw water, which could be attributable to the presence of bromine as a contaminant in sodium hypochlorite (the disinfection agent). Water quality tests carried out by the water treatment plant (WTP) and by the State Sanitary Inspectorate after the modification of the process line confirmed the high quality of water in the distribution network after the change of disinfection method.

**Keywords:** water treatment plant; disinfection by-products; ozonation installation; bromate

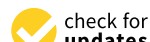



## 1. Introduction

Public health and welfare rely on access to water that is free from microbiological and chemical pollutants, so the quality of ground water and surface water to be used as drinking water has to be monitored and rigorously tested [1,2]. Another vital aspect is the elimination of secondary pollution with biological contaminants (bacteria re-growth) in industrial water supply lines. The introduction of water disinfection processes is considered to have been a major breakthrough for mankind and the key factor in the reduction of mortality rates. Growing public awareness stimulated, inter alia, by advances in analytical methods, has caused the rising demand for better quality water—both in terms of micro-biological, physico-chemical and organoleptic parameters [2,3].

The predominant and widely applied disinfection agent is still chlorine, mostly used in the form of sodium hypochlorite or chlorine dioxide, though water disinfection with ozone is becoming increasingly popular. Additionally, new technologies are being tested [4–7].

The first attempts at water disinfection with ozone date back to 1906 in France. In Kraków, ozone was first used in 1956 to treat water drawn from the Vistula River. In the 1970s, seven water treatment plants (WTPs) in Poland used ozonation treatment [8]. Ozone disinfection is the most reliable and effective treatment against pathogenic microorganisms, such as Giardia lamblia. Numerous cases of infections registered in the USA caused by

Giardia lamblia cysts, alongside the unfeasibility of completely eliminating this biohazard in water distribution networks, have prompted researchers' interests in ozonation technology [9,10]. Apart from the microbiological aspect, the main advantage of ozonation lies in the improvement of the taste and odor (organoleptic parameters) of tap water, thus improving customers' satisfaction [9,11,12].

Disinfection by-products (DBPs) are formed as a result of the oxidation of natural organic matter in a water environment and in reactions with certain microcomponents, such as bromides or iodides. According to reports in the literature, there are over 600 such chemical compounds, whilst less than 100 of these are considered a potential health concern [13,14]. Among chlorination by-products are trihalomethanes (THMs), haloacetonitriles, haloacetic acids, haloaldehydes, haloketones and chlorophenols. The by-products of water treatment with ozone include bromates, low-molecular-weight organic acids e.g., carboxylic acids, aldehydes, ketones, bromate-containing organic compounds, bromated THMs, bromoacetic acids and hitherto unidentified yet readily degradable oxidized polar organic compounds [13,15].

The DBPs for which adverse health effects have been verified include THMs and bromates. THMs were recognized as substances of significant health concern in the 1960s. In the International Agency for Research on Cancer (IARC) classification, chloroform ($CHCl_3$) and bromodichloromethane ($CHBrCl_2$) are categorized as Class 2B carcinogens, i.e., potentially carcinogenic to humans [16]. Their maximal concentrations in drinking water need to be rigorously controlled. In accordance with regulations currently in force in Poland and in other countries worldwide, the maximal concentrations of $CHCl_3$, $CHBrCl_2$, $CHBr_2Cl$ (chlorodibromomethane) and $CHBr_3$ (bromoform) must not be higher than 100 µg/L. Further, $CHBrCl_2$ content must not exceed 15 µg/L, whilst $CHCl_3$ concentration must not be higher than 30 µg/L [17].

Major determinants of the disinfection process and, consequently, the amounts of thus formed THMs include the actual oxidizer dosage, contact time, temperature and raw water properties, such as natural organic matter contents, concentrations of bromides, iodides, and ammonium ions and the pH conditions [13,15,18–21].

In 1990, bromates were categorized as potential human carcinogens and, in 1993, the WHO recommended that bromate contents should be rigorously controlled. The maximal allowable concentration of $BrO_3^-$ ions in drinking water must not be higher than 10 µg/L, and the WHO recommends that it should be kept as low as possible. Bromides can be directly oxidized to bromates by molecular ozone, or in an indirect process via radical reactions (multi-phase processes). The first product of the reaction of ozone molecules with bromides is hypobromite, an intermediate phase. The reaction of hypobromites with ozone yields bromides and bromates [22,23]. Similar to THMs, the formation of bromates is controlled by the actual ozone dose, temperature, pH conditions and the bromide concentrations in source waters [18,23–26]. Depending on the above critical factors, a fraction (5–30%) of bromides will form bromates [27]. It has been reported that $Br^-$ ion concentrations in excess of 0.05 mg/L may lead to the formation of excessive amounts of $BrO_3^-$ [22]. The role of critical factors controlling the formation of DBPs has received a great deal of attention recently. Extensive research efforts have continued, including laboratory-scale testing, pilot plant testing, investigations carried out at water treatment works [28–35], and local reports [15,36–39]. The mutagenicity of chlorinated water is higher than that of water treated with minimal doses of chlorine and ozone [40,41]. THM concentrations in public drinking water systems are rigorously controlled and kept below 50 µg/L [32,42]. In certain cases, however, levels in excess of 100 µg/L have been reported [30,33,37]. High concentrations of DBPs are of major concern, particularly in developing countries where water resources are limited [20,38,43]. Studies of DBPs, the physico-chemical parameters of water and of disinfection agents will be used to effectively model the processes of their formation. Most models reported in the literature have relevance to THMs formation [18,20,35,44,45]. However, in some cases, laboratory-scale models prove inadequate in real water treatment plant (WTP) conditions. In the case of

bromates, the problem lies in radical reactions which are controlled by the local properties of waters and the water treatment strategy [25,46].

Generally, the presence of bromates in water is mostly attributed to the disinfection with ozone even though ozonation may not be the decisive factor. Actually, in some cases, bromates are reported to have been detected in waters treated without the use of ozone, and in processes where source waters were bromide-free, which may be due to sodium hypochlorite contamination with bromide compounds [36,47–50].

One of the side-effects of water treatment with ozone is the occurrence of carbonyl compounds, including low-molecular-weight carboxylic acids, aldehydes and ketones. They emerge following the cleavage of a double-bond C = C. Aldehydes can also be formed in a reaction of organic matter with chlorine dioxide and chlorine [51]. Formaldehyde has been categorized as a potential human carcinogen, yet the WHO has not suggested a value of the maximal admissible concentrations of formaldehyde in drinking water, because formaldehyde normally occurs at concentrations well below those of health concern [52,53].

Local water treatment plants (WTPs) treating low-quality source waters need to apply high doses of disinfection agents. To reduce chlorine dosage, the ozonation step is introduced concurrent to chlorination at the end of the water treatment line. Another way to reduce the dosage volume is in-line chlorination. In this case, installing online water quality sensors can be helpful in monitoring water quality [5,7].

Ozonation still remains a widely applied water treatment method, though it usually requires the upgrading of the water treatment line. Ozonation is usually not performed simultaneously with chlorination, as described in this paper. Here a strategy is proposed whereby the disinfection process was modified through the use of a mobile installation. The application of a small-size, fully automated ozonation installation only slightly complicated the water treatment process, without the need to redesign the water treatment line, and with relatively low investment costs. Additionally, the installation can be easily connected to or disconnected from the process line. The aim of this paper is to explore the potential of water disinfection with ozone as a strategy to minimize chlorine doses, using a water treatment plant in Skawina (WTP Skawina) as the case study. The effectiveness of the modified water disinfection scheme and quality of the treated water were verified through extensive testing programs involving measurements and monitoring of concentrations of selected disinfection by-products. The goal of this study was to analyze whether a change of disinfection method affects the final water quality. The results of water quality testing conducted when the disinfection system was modified were collated and compared. Some water quality tests and laboratory analyses performed within the framework of our research are not part of the standard test procedures used at water treatment plants and therefore have not been presented in publications on water treatment in Poland. These included, inter alia, measurements of total bromine and bromide. The ultimate goal of this study was to establish whether the process is likely to prompt the formation of bromate and main bromine disinfection by-products. To date, total bromine and bromide levels have not been routinely determined at water treatment plants in Poland, which is why the results cannot be directly related to past data from the water treatment plants in Skawina or elsewhere.

## 2. Materials and Methods

### 2.1. Water Treatment System at the WTP Skawina

Skawina, a small town with 30 thousand inhabitants, is located in the southern part of Poland, 15 km south-west of Kraków (Figure 1). The WTP in Skawina with a maximum capacity of 500 $m^3$/h supplies water to 3450 users in Skawina and in the nearby area. Water supplied to the public drinking water system comes from three types of water intake (surface water intake, infiltration water facility and deep wells). The main water supply comes from the infiltration water intake in the proximity of the Skawinka River and from an emergency surface water intake. The Skawinka River, a tributary of the Vistula River, is 33 km long with a basin area of 352.4 $km^2$.

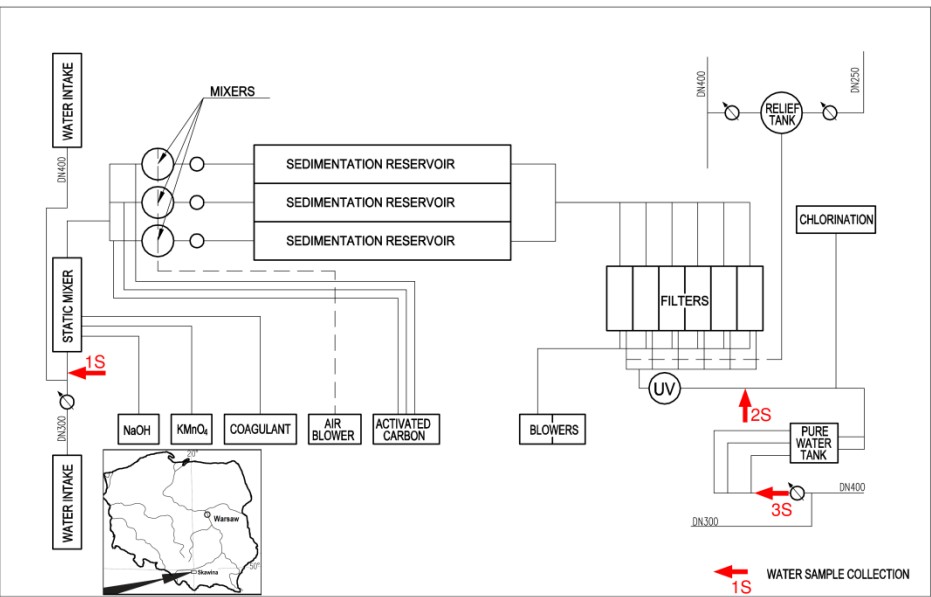

**Figure 1.** Water treatment line at WTP Skawina.

The surface water intake facility is located at the bottom of the Skawinka River, 5.5 km from the point where it flows into the Vistula River. It has two sections: water drains and boxes made from reinforced concrete with a grille. The surface water intake is connected to the WTP Skawina via a pipeline Ø 400 mm. The capacity of the water intake facility is 400 m$^3$/h.

The surface water intake comprises two systems of wells (Starorzecze 1 and Starorzecze 2) located within the long-abandoned river-bed of the Skawinka River. The intake facility Starorzecze 1 comprises 4 deep wells 12 m in depth and 1200 mm in diameter, provided with horizontal drains (2 drainpipes in each well) and a pumping engine. The intake facility Starorzecze 2 consists of 7 wells provided with drains. Water flows into one receiving collector well where it is supplied to the WTP. The total length of all drainpipes in the infiltration water intake facility is 500 m. Within the abandoned river-bed there is a water reservoir 40,000 m$^3$ in volume surrounded by wells. The infiltration of waters from the reservoir further contributes to the well capacity. This reservoir provides additional water reserves during elevated water turbidity periods. The total capacity of the infiltration water intake facility is 300 m$^3$/h.

Water from the intake facilities is supplied to the WTP Skawina in Radziszowska Street via two independent pipelines Ø 400 mm and Ø 300 mm (Figure 1). Water admitted to the static mixer is then admixed with a 30% solution of NaOH, 3% solution of KMnO$_4$, coagulating agent PAXXL10 (aqueous solution of polyaluminum chloride) and powdered activated carbon.

Water admitted to the static mixer is then admixed with a 30% solution of NaOH, 3% solution of KMnO$_4$, and coagulating agent PAXXL10 (aqueous solution of polyaluminum chloride), and then the water flows to the rapid mix where it also aerated. From the rapid mix via a low-rate mix, water flows down to the open-air longitudinal settling tanks. The retention time in settling tanks ranges from 4 to 6 h, and afterwards water flows through high-rate filters. There are 6 filtration chambers at the WTP Skawina; filters are made of sand layers and anthracite, and each is rinsed separately to ensure the continuous operation of the WTP. The final step in the treatment line is water disinfection using the UV irradiation system (6 medium-pressure lamps) and chlorination. The WTP Skawina uses a medium-pressure UV-C Berson incorporating 6 UV lamps with power ratings of 2200 W each. The disinfection agent added to water is sodium hypochlorite (6–9 g/m$^3$), with 18% active chloride contents. After disinfection, water flows to the contactor where it is directly supplied to the public water distribution system via pressure pumps, whilst the

excess water is collected in equalizing tanks, positioned at an altitude of 270 m above the sea level, which is 50 m above the WTP site to ensure the sufficient pressure levels in the water distribution system at the instant the pumps at the WTP Skawina are turned off [54].

*2.2. Raw Water Quality at WTP Skawina*

The main water intake in the WTP in Skawina comprises the surface water and infiltration water intake facilities, both receiving water from the Skawinka River where the water quality is described as inadequate. In terms of biohazards, the water in the Skawinka River has been categorized as Class 3 water; in terms of its hydromorphology it is classified as Class 2 water. In regards to its physico-chemical parameters, the water has been classified as potentially less than good. Specifically, the overall quality of water in the Skawinka River has been described as threatened or impaired [55]. In terms of microbiology, water in the Skawinka River upstream of the town of Skawina is categorized as Class A3 water, mostly due to the presence of Escherichia coli and fecal coliforms. The results of microbiological tests conducted by the State Sanitary Inspectorate are collated in Figure S1 (Supplementary Materials). Monitoring and tests conducted by the Health Inspectors in 2013 and 2014 and in earlier years also revealed the presence of fecal coliforms. Class A3 describes the water quality as threatened and impaired in the context of requirements for surface waters to be used for human consumption. Class A3 waters require high-efficiency reliable physical and chemical treatment systems, including oxidation, coagulation, flocculation, filtration, activated carbon filtering and disinfection, involving the ozonation and final chlorination steps [56].

Ground water intake facilities in Skawina draw water from a quaternary aquifer.

Water quality is described as threatened or even impaired, which is mostly due to the ground water source being a little beneath the surface, due to the absence or inadequacy of the aquifer insulation as well as the occurrence of numerous sources of pollution [57]. Ground water quality categorized as satisfactory and surface water quality described as threatened present a major challenge to the WTP Skawina as they need to supply water that meets the stringent criteria for drinking water quality. The quality of water supplied to the inhabitants of Skawina and the adjacent areas has systematically improved through upgrading the water treatment facilities and operational practices.

*2.3. Ozonation Step as a Modification to the Water Disinfection Process*

The water treatment line at the WTP Skawina has been modified to minimize the doses of chlorine applied following the UV radiation step. When water is treated with chlorine or chlorine compounds, the concentrations of free chlorine in water (at the water-access points) must not exceed 0.3 mg/L. The admissible maximal dose for ozone is 0.05 mg/L [17]. The disinfection process was modified through adding a mobile system generating and dosing ozonated water, following the UV irradiation step. The disinfection process involving chlorination and ozonation was employed from November 2017 through January 2018 and in May and June 2018. Throughout the period February–May 2018, water was treated using a traditional process, with the use of sodium hypochlorite.

The mobile ozonation system manufactured by Wofil is an autonomous installation producing ozonated and degassed water to be added to the water being treated. Ozone used to prepare ozonated water is collected from oxygen in the air, with the use of ozone generators having the maximum capacity of 70 g $O_3$/h. The nominal capacity of the entire system is 4 $m^3$/h of water ozonated under the pressure of 12 bar. Interestingly, this technology relies exclusively on natural processes. Ozone concentrations in highly ozonated waters can vary from 0.1 to 3 ppm. The optimal performance of the ozonation system is achieved when the ozone concentration is 1.5 ppm. The ozonation system requires electric power and water supply. After disinfection, ozone decomposes, forming pure oxygen, and the elements of the ozone installation do not require cleaning or rinsing, so the water consumption can be thus minimized.

The ozonation system (Figure 2) incorporates a generator producing ozone from atmospheric air, contact columns generating ozonated water with the required concentration and a de-aeration installation to minimize the desorption of residual ozone in the public water supply system.

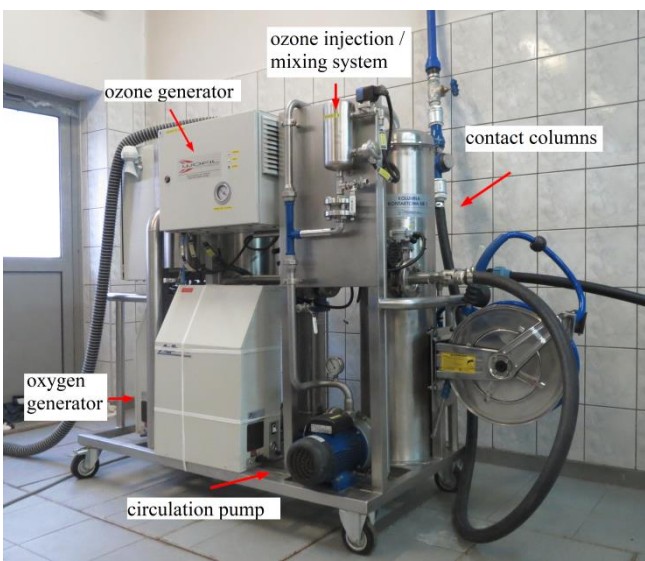

**Figure 2.** Mobile ozonation system.

The main components of the ozone installation are sensors detecting the presence of residual ozone in water and in the air, spray-free cleaning nozzles and elements triggering the decomposition of residual ozone. Water is supplied via a pipeline; flow control is effected through the use of a manually controlled valve and an electric-drive throttling valve which automatically admits water to the 1 contact column. Contact columns are cylinder-shaped and flat-bottomed components, 250 mm in diameter and 140 cm in height. The total volume of the two columns is approximately 0.14 m$^3$. Columns made of stainless-steel resistant to ozone action are equipped within the system, preventing uncontrolled gas release and enabling the removal of excess water in the case of an overflow. When the water level is raised or lowered, the sensor based on pressure transducers causes the flow control valve supplying water to contact columns to be automatically opened or closed. Excess gas desorbing from water in the form of a mixture of oxygen and ozone is degassed in the other contact column and then pushed outside by a fan triggering ozone decomposition. The main purpose of the installation is to feed ozonated water to water being treated, in order to achieve the following goals:

- To enhance the bacteriostatic action within the public water supply system;
- To minimize the dosage of chlorine added to water;
- To remove any odors remaining after the water treatment processes;
- To ensure a better taste of water supplied to the end users.

Furthermore, the ozonation installation can effectively interact with the public water supply system or can be incorporated at other points within the water treatment train for the disinfection, rinsing and removal of contaminants.

The ozonation installation is compact, 60 cm in width, 170 cm in height and 200 in length; its design enables its transportation on standard transport vehicles, such as a bus. The system is fixed on a frame provided with rails and wheels for easy handling and transport. The main advantage of the ozonation system is that the ozonation step can be easily incorporated in small treatment plants (400 m$^3$/h capacity) and medium-size WTPs (with capacities in the range 1500–200 m$^3$/h), with little investment costs. The ozonation system does not require any dedicated infrastructure or highly qualified engineers for its operation and maintenance. Its structural design and process parameters are such that it

can be connected to the water network with a capacity of up to 400 m$^3$/h. Other models are now available to handle lower or significantly higher flow rates.

*2.4. Water Quality Monitoring at the WTP Skawina Prior to and after Modifications of the Disinfection Process*

The WTP Skawina is a medium-sized WTP. The structure of the water distribution systems in Poland and the results of inspections carried out by the State Sanitary Inspectorate have revealed that the current water quality standards are seldom exceeded and, if so, this mostly occurs in smaller plants. Among WTPs similar in size to the WTP Skawina, the water quality criteria were met in 99% of them [58]. Studies on upgrading the treatment lines at WTPs are aimed at improving the reliability and efficiency of disinfection processes. It has been demonstrated that modifications of the operational practices have resulted in reduced concentrations of DBPs [12,32,33].

Water supplied to the public water distribution system after disinfection using sodium hypochlorite satisfies the microbiological requirements, yet in some cases excessive concentrations of free chlorine have been reported (Figure S2-Supplementary Materials).

The chlorine concentrations varied from 0.1 to 0.6 mg/L; the regulatory level is 0.3 mg/L. There have been complaints from customers not satisfied with the organoleptic parameters of water. The admission of ozone to water supplied to the public water distribution system and concurrent minimization of the chlorine contents aimed to improve the taste and smell of water, at the same time removing the risks to public health, particularly those of a microbiological nature.

The effects of the ozonation step on water quality were evaluated through measurements and the monitoring of selected physico-chemical parameters of treated water, including the concentrations of DBPs. The monitoring program involved three steps. Water samples were collected on 3 January 2018, 8 May 2018, and 5 June 2018. In the first and third steps, water samples were investigated, whilst the mobile ozonation installation was operational. In the second step, water samples were collected when the disinfection was carried out with sodium hypochlorite only. Water samples were collected at three points on the process line (Figure 1) (1S—raw water; 2S—water after filtration and UV radiation; and 3S—water after disinfection), to be supplied to the public water system. As the amounts of thus formed DBPs are controlled by the precursor–oxidiser contact time [18,20], in steps 2 and 3, samples were also collected at users' homes (Ra) (at a distance of about 8 km from the WTP).

Alongside the tests carried out during the upgrading of the water treatment systems, the results of tests conducted by the State Sanitary Inspectorate prior to and after the system modification were taken into consideration.

Water samples were collected in accordance with the obligatory procedure and the physico-chemical parameters of water were determined by the recommended reference methodology [59,60]. Testing of a raw water sample collected on 3 January 2018 consisted of measurements of electrical conductivity (EC), pH values, bromate concentrations, total bromine, total organic carbon (TOC), formaldehyde and bromates ($BrO_3^-$). From the absorbance measurements we obtained values of $\alpha$ = 254, 272, and 436 nm. The same parameters were determined in samples 2 and 3 and THM concentrations were measured in the water sample collected after disinfection. Alongside all previously listed parameters, ion concentrations ($Na^+$, $Ca_2^+$, $Mg_2^+$, $Fe_2^+$, $HCO_3^-$, $Cl^-$, $NH_4^+$, $NO_3^-$, $NO_2^-$ and dissolved organic carbon (DOC)) were determined in the water samples collected during the spring. Measurements of $BrO_3^-$ were taken in water samples collected on 8 May 2018 and 5 June 2018. On 5 June 2018, measurements were taken at two independent laboratories to determine the THM concentrations.

The physico-chemical parameters of water were investigated by the pH-potentiometry, EC was measured by the conductometric method and total bromine contents were determined by inductively coupled plasma-mass spectrometry (ICP-MS). $Ca_2^+$, $Mg_2^+$, $HCO_3^-$ and $Cl^-$ contents were determined by titration methods; $Na^+$ and $K^+$ concentrations were

established by atomic absorption spectrometry (ASA), $Br^-$, $NO_3^-$, $NO_2^-$, $NH_4^+$ and $BrO_3^-$ measurements were measured using ion chromatography. Absorbance was measured by the spectrophotometry method UV-VIS; TOC and DOC were established by IR spectrometry. THM concentrations were determined by gas chromatography methods; $Fe_2^+$ and formaldehyde levels were obtained by the colorimetric method.

In this study, water quality was compared between two seasons (winter and spring). Water was treated using two alternative treatment schemes: chlorination and ozonation plus chlorination. Analyses were performed that were not part of the standard testing procedures at water treatment plants, to determine the concentrations of compounds, showing the presence of organic matter (TOC, DOC, and absorbance) and of total bromine and bromates in water samples collected at various stages of the water treatment process.

Water samples obtained from two different treatment processes were compared and water parameters were closely monitored and registered at all stages of the treatment process (from raw water, after filtration and after disinfection involving chlorination or chlorination plus ozonation).

## 3. Results and Discussion

Throughout the duration of the testing program, the ozonation plant was systematically monitored to register the amounts of residual ozone immediately before dosing in the public water system and in the water distribution network supplying the town. The ozonation system operated at 65–90% of its nominal capacity. Taking into account the duty regime of the ozonation system and the water flow rates to be handled, the ozone doses were derived accordingly whilst the actual doses of ozone and chlorine admitted to the water distribution network were measured in the pumping station and, in special cases, also at selected points in the water distribution network. Chlorine and ozone concentrations registered in water after disinfection and collected in the pumping station whilst the ozone installation was operational in the winter were: chlorine: 0.029–0.041 mg/L; $O_3$: 0.004–0.25 mg/L. In spring, the chlorine concentration ranged from 0.2 mg/L to 0.47 mg/L, whilst the $O_3$ concentration remained constant (0.05 mg/L). Variable ozone dosages and chlorine and ozone concentrations in disinfected water supplied to the public water supply system are collated in Figure 3.

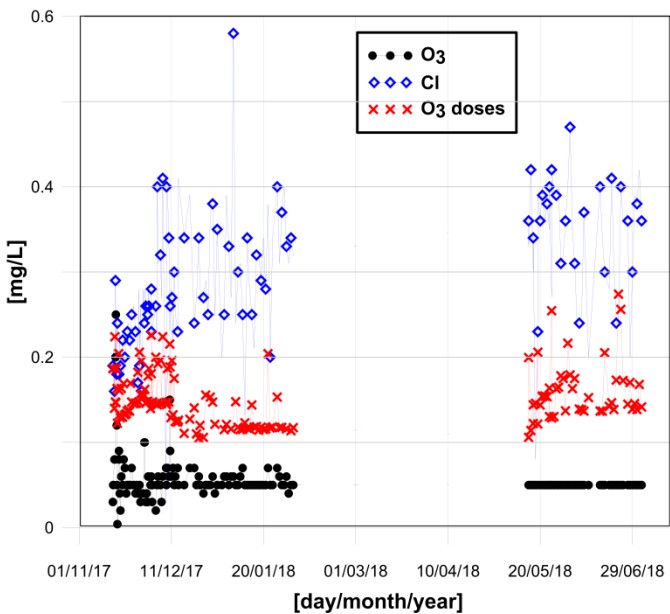

**Figure 3.** Chlorine and ozone concentration in water after disinfection.

After the introduction of the ozonation step, water quality was described as very good and its organoleptic parameters were vastly improved. In terms of physico-chemical and

microbiological parameters (Table S1), the water treated at the WTP Skawina was found to be fit for human consumption. These findings were compared with results of earlier monitoring programs launched at the WTP Skawina and using samples collected when water was treated with sodium hypochlorite only.

The physico-chemical parameters of water disinfected with chlorine and ozone are summarized in Tables 1–3.

**Table 1.** Selected physico-chemical parameters of water from the WTP Skawina treated with ozone (sample collected on 3 January 2018).

|  | Sample 1S | Sample 2S | Sample 3S |
|---|---|---|---|
| pH | 7.01 | 7.61 | 7.4 |
| EC [mS/cm] | 0.452 | 0.473 | 0.478 |
| Absorbance ($\alpha = 254$) | 0.071 | 0.043 | 0.038 |
| Absorbance $\alpha = 272$) | 0.038 | 0.032 | 0.026 |
| Absorbance ($\alpha = 436$) | 0.008 | 0.007 | 0.001 |
| Formaldehyde [µg/L] | <6 | <6 | <6 |
| $Br_{total}$ [µg/L] | 18 | 16 | 89 |
| $Br^-$ [µg/L] | 22.5 | 22.7 | <20 |
| $BrO_3^-$ [µg/L] | <2 | <2 | <2 |
| TOC [mg/L] | 1.79 | 1.58 | 1.54 |
| $CHCl_3$ [µg/L] |  |  | 6.5 |
| $CHBrCl_2$ [µg/L] |  |  | 2.5 |
| $CHBr_2Cl$ [µg/L] |  |  | <2 |
| $CHBr_3$ [µg/L] |  |  | <2 |
| $\Sigma THM$ [µg/L] |  |  | 9.0 |

**Table 2.** Selected physico-chemical parameters of water from the WTP Skawina treated with chlorine exclusively (sample collected on 8 May 2018).

| | Sample 1S | Sample 2S | Sample 3S | Ra (Radziszów) | Parametric Value * | Standard Analytical Methods |
|---|---|---|---|---|---|---|
| pH | 7.16 | 7.18 | 7.19 | 7.28 | 6.5–9.5 | PN-EN ISO 10523:2012 |
| EC [mS/cm] | 0.532 | 0.544 | 0.547 | 0.544 | 2.5 | PN-EN 27888:1999 |
| Absorbance $\alpha = 254$ | 0.104 | 0.042 | 0.029 | 0.030 |  | PN-C-04572,1984 |
| Absorbance $\alpha = 272$ | 0.093 | 0.034 | 0.026 | 0.026 |  | |
| Absorbance $\alpha = 436$ | 0.016 | 0.000 | - | - |  | |
| $Na^+$ [mg/L] | 23.40 |  | 26.43 | 27.07 | 200 | PN-ISO 9964–2 1994 |
| $K^+$ [mg/L] | 4.96 |  | 4.996 | 4.784 | - | |
| $Ca^{2+}$ [mg/L] | 60.12 |  | 76.15 | 80.16 |  | PN-ISO 6058,1999 |
| $Mg^{2+}$ [mg/L] | 19.46 |  | 12.04 | 13.13 | 7–125 | PN-ISO 6059,1999 |
| $Fe^{2+}$ [mg/L] | 0.56 |  | <0.2 |  | 0.2 | PN-ISO 6332,2001 |
| $Cl^-$ [mg/L] | 49.64 |  | 48.87 | 42.55 | 250 | PN-ISO 9297,1994 |
| $HCO_3^-$ [mg/L] | 27.46 |  | 30.51 | 27.46 |  | PN-EN ISO 9963–1, 2001 |
| $NH_4^+$ [mg/L] | 0.19 |  | 0.22 | 0.20 | 0.5 | PN-EN ISO 14911:2002 |
| $NO_3^-$ [mg/L] | 4.6 |  | 4.8 | 4.9 | 50 | PN EN ISO 10304–1:2009 |
| $NO_2^-$ [mg/L] | <0.01 |  | <0.01 | <0.01 | 0.5 | |
| TOC [mg/L] | 2.94 | 2.18 | 2.31 |  |  | PN-EN 1484, 1999 |
| DOC [mg/L] | 2.37 | 1.88 | 2.01 |  |  | |
| Formaldehyde [µg/L] | 13 | 10 | 11 |  |  | PB-W-11 |
| $Br_{total}$ [µg/L] | 60 | 51 | 165 | 150 |  | PN-EN ISO 17294 |
| $Br^-$ [µg/L] | 29.6 | 28.1 | <20 | <20 |  | PN-EN ISO 10304–1:2009 |
| $BrO_3^-$ [µg/L]   Lab.1 | <5 | <5 | <5 | <5 |  | DIN EN ISO 15061 |
| Lab.2 |  |  | 7.9 | 6.6 | 10 | |
| $CHCl_3$ [µg/L] |  |  | 12.6 | 16.7 | 30 | |
| $CHBrCl_2$ [µg/L] |  |  | 2.8 | 6.0 | 15 | |
| $CHBr_2Cl$ [µg/L] |  |  | <2 | <2 |  | PN-EN ISO 10301:2002 |
| $CHBr_3$ [µg/L] |  |  | <2 | <2 |  | |
| $\Sigma THM$ [µg/L] |  |  | 15.4 | 22.7 | 100 | |

Note: * according to Journal of Laws, 2017 item 2294 [17].

The values of all investigated parameters were found to be well below the parametric value recommended for drinking water. Testing conducted on ozonated water samples did not reveal any determinable amounts of bromates. Measurements of $BrO_3^-$ concentrations were repeated twice during winter and twice in spring, and in one case (see Table 3) were taken by two independent laboratories. It is worth mentioning that THM concentrations in water collected at the WTP Skawina and at customers' homes in the period prior to the

introduction of the ozonation step were in some cases higher than those registered when ozonation was introduced as the additional disinfection step (Figure 4).

**Table 3.** Selected physico-chemical parameters of water from the WTP Skawina treated with ozone (sample collected on 5 June 2018).

|  |  | Sample 1S | Sample 2S | Sample 3S | Ra (Radziszów) |
|---|---|---|---|---|---|
| pH | | 7.28 | 7.25 | 7.27 | 7.30 |
| EC [mS/cm] | | 0.515 | 0.513 | 0.520 | 0.554 |
| Absorbance ($\alpha$ = 254) | | 0.172 | 0.080 | 0.065 | 0.042 |
| Absorbance $\alpha$ = 272) | | 0.149 | 0.065 | 0.048 | 0.030 |
| Absorbance ($\alpha$ = 436) | | 0.036 | 0.006 | 0.001 | 0.001 |
| $Na^+$ [mg/L] | | 25.51 | | 29.66 | 30.98 |
| $K^+$ [mg/L] | | 5.65 | | 5.55 | 5.62 |
| $Ca^{2+}$ [mg/L] | | 58.52 | | 54.78 | 62.79 |
| $Mg^{2+}$ [mg/L] | | 8.76 | | 8.92 | 4.86 |
| $Fe^{2+}$ [mg/L] | | 0.75 | | <0.2 | |
| $Cl^-$ [mg/L] | | 46.09 | | 56.73 | 63.82 |
| $HCO_3^-$ [mg/L] | | 24.41 | | 27.46 | 25.93 |
| $Ca^{2+-}$ [mg/L] | | 58.52 | | 54.78 | 62.79 |
| $Mg^{2+-}$ [mg/L] | | 8.76 | | 8.92 | 4.86 |
| $NH_4^+$ [mg/L] | | 0.35 | $0.063 \pm 0.011$ | <0.015 | - |
| $NO_3^-$ [mg/L] | | 3.6 | $4.2 \pm 0.5$ | $4.2 \pm 0.5$ | - |
| $NO_2^-$ [mg/L] | | 0.21 | - | - | - |
| TOC [mg/L] | | 4.26 | 3.38 | 3.32 | - |
| DOC [mg/L] | | 3.71 | 3.34 | 3.07 | - |
| Formaldehyde [µg/L] | | 11 | <6 | 14 | - |
| $Br_{total}$ [µg/L] | | 31 | 31 | 46 | 58 |
| $Br^-$ [µg/L] | | 25.2 | 25.6 | <20 | <20 |
| $BrO_3^-$ [µg/L] | Lab.1 | | | <2 | <2 |
| | Lab.2 | | | <5 | <5 |
| $CHCl_3$ [µg/L] | Lab.1 | | | 18.4 | 17.5 |
| | Lab.2 | | | 16.2 | 14.6 |
| $CHBrCl_2$ [µg/L] | Lab.1 | | | 2.6 | 6.2 |
| | Lab.2 | | | 2.36 | 5.13 |
| $CHBr_2Cl$ [µg/L] | Lab.1 | | | <2 | <2 |
| | Lab.2 | | | <1 | 1.31 |
| $CHBr_3$ [µg/L] | Lab.1 | | | <2 | <2 |
| | Lab.2 | | | <1 | <1 |
| $\Sigma THM$ [µg/L] | Lab.1 | | | 21.1 | 23.7 |
| | Lab.2 | | | 18.2 | 21.1 |

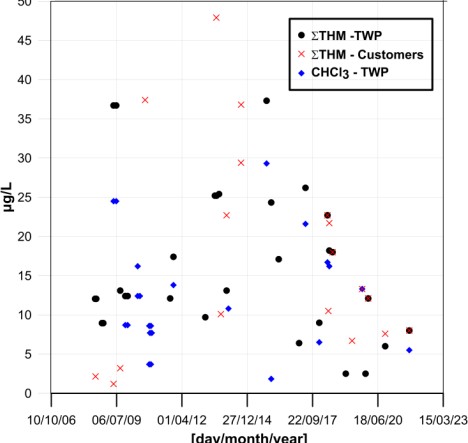

**Figure 4.** THM concentration in treated water (data provided by the State Sanitary Inspectorate, the Health Inspectorate, Water Works in Kraków [61] and test results—Tables 1–3).

Raw water collected in winter and in spring contained different amounts of total organic carbon and their absorbance values were also different. These parameters are excellent indicators of the organic matter contents in water and are widely applied when evaluating the effectiveness of organic matter removal in process lines [35,62–66]; hence, they could be helpful when assessing the potential of DBP formation [67]. The registered values of these parameters are indicative of lower organic matter contents in the winter. Regardless of the actual raw water quality, after the filtration step in the water treatment train, the absorbance value tended to decrease ($\alpha = 254$) by nearly 40% in relation to the initial level, which confirms the reliability and efficiency of the adopted water treatment method. The removal of organic matter is a critical factor reducing the risk of formation of DBPs (particularly THMs) within the water distribution system. The difference in UV absorbance at a wavelength of 272 nm before and after chlorination has been found to be linearly proportional to the concentration of many DBPs [66]. In this case, due to the small numbers of samples, the values of absorbance at a wavelength of 272 nm (Tables 1–3) give us only some indication of the processes involved. In our experiments, samples (2S, 3S) were collected almost at the same time.

Samples collected in the spring revealed the presence of formaldehyde, the concentrations of which changed in the subsequent stages of the treatment process; this tendency has also been observed in other WTPs [68]. Seasonally varying trace amounts of formaldehyde are present in the water environment [51]. Consequently, formaldehyde concentration levels detected in facilities treating mostly surface waters range from amounts below the threshold of detection to well over 100 µg/L [68–70]. Because of varied atmospheric conditions (rainfall intensity and temperature), the highest concentrations of formaldehyde are registered in spring and the lowest in winter. Ozonation as well as chlorination affects aldehyde concentrations [69,71,72]. Coagulation/flocculation and sand filtration enable the removal of 64–80% of aldehydes and the application of granular activated carbon helps remove a further 15–64% [68]. The formaldehyde concentration in the water tested in June was found to decrease after the coagulation and filtration process. After the disinfection process, the concentration of formaldehyde increased (Table 3).

Throughout the entire testing program (steps 1, 2 and 3):

- Free bromine and no bromates were found in water samples after disinfection (3S) and in water samples collected at users' homes (Ra);
- Elevated total bromine contents in relation to samples 1S and 2S were detected in samples after disinfection (3S and Ra) (Figure 5);
- In raw water samples (1S) and in water collected before disinfection (2S), bromine ions and total bromine contents were at similar levels, which indicates that bromine will occur in the form of ions in water.

The comparison of bromides and total bromine contents (1S, 2S) suggests that bromine is present in raw waters in the form of bromide compounds, though bromine-containing organic matter is likely to occur as well. The amounts of bromine in water samples collected in the spring are larger than in winter. The seasonal variability of bromine concentrations in surface water is addressed in previous research [73]. The higher concentrations registered in the spring may be attributable to surface runoffs [74,75].

Bromide concentration is a decisive factor controlling the type and amounts of thus formed DBPs [27,76]. Bromo-organic compounds emerging in the presence of bromide as DBPs are regarded as potential genotoxic carcinogens and are more hazardous than their chlorine-containing equivalents [13].

Elevated total bromine concentrations detected in samples (3S and Ra) in relation to 1S and 2S can be attributed to the contamination of sodium hypochlorite with bromine. Elevated bromate concentrations in sodium hypochlorite are associated with the bromide contents in salt (NaCl) [36,47–50]. Examples of bromate concentration levels in sodium hypochlorite and in treated waters are collated in Table S2 (Supplementary Materials).

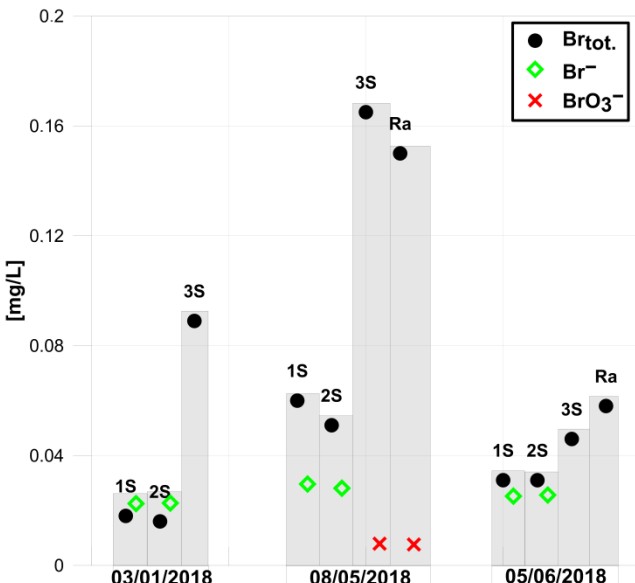

**Figure 5.** Concentrations of bromine, bromide and bromates in raw (1S) water, after filtration and UV radiation (2S), after disinfection (3S) and at users' homes (Ra).

During the disinfection with chlorine, bromide ions react with hypochlorous acid and are oxidized, forming hypobromous acid (HBrO) or hypobromite. In the range of neutral pH levels, the amounts of emerging $HBrO/BrO^-$ are at least five times larger than those of the remaining bromine compounds [76]. Non-dissociated forms of HBrO will react more readily with organic matter, leading to the formation of bromo-organic chlorination by-products [22]. The occurrence of non-dissociated HBrO in sample 3S cannot be entirely precluded, considering the fact that the amounts of emerging $Br^-$ and $BrO_3^-$ ions were below the threshold of detection. In samples 3S and Ra, bromine could be present in the form of other, hitherto unidentified bromine compounds.

The presence of bromates was detected in waters treated exclusively with chlorine, through testing conducted by an accredited laboratory (2) (Table 2). These findings alongside the elevated concentrations of total bromine in disinfected water are suggestive of sodium hypochlorite contamination with bromine as the potential source of bromates. This issue, however, requires further research.

The effectiveness of the modified water disinfection process was studied by analyzing the variability of TOC concentration levels and adsorbance ($\alpha254$ and $\alpha272$) in raw water samples (1S), in water after filtration (2S) and after the disinfection step (3S) (Figure 6). Apparently, the TOC concentration tends to decrease after filtration and so does the total bromine content. Disinfection with sodium hypochlorite leads to a rapid increase in total bromine concentration (Figure 6a). The difference between the total bromine concentrations in water samples 3S and 1S was most significant in the case of waters subjected to chlorination only. The most significant variability of absorbance of water samples $\alpha254$ and $\alpha272$ (after the final treatment 3S and raw water 1S) was registered in the case of the ozonation and chlorination process (Figure 6b), which is suggestive of an improved water treatment performance.

The test results obtained during the first stage of the monitoring program (winter) showed that the introduction of ozone did not result in bromate formation despite the presence of bromide ions in raw water. Preliminary test data were confirmed by further results. In the spring, bromate concentration measurements were repeated twice; in one case, the measurements were taken by two independent laboratories (Table 3). Nevertheless, the long-term monitoring of bromate concentrations would contribute towards answering the original research question on how DBPs are actually formed. Determining the concentrations of other bromine compounds (including brominated organic compounds) is likely to help account for discrepancies between the total bromine and bromide concentrations

in finished waters. In the conditions of WTPs handling water for which the quality is described as low and subjected to seasonal changes, monitoring programs are of great importance in the context of the potential optimization of water treatment practice and in regard to public health protection.

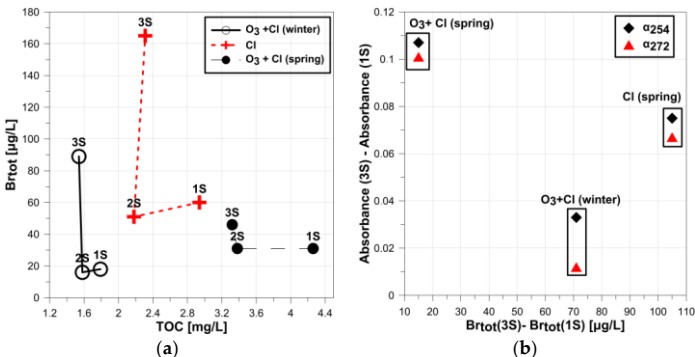

**Figure 6.** Parameters of water in the water treatment line (TOC, absorbance and total bromine concentration in raw water (1S), water after filtration (2S) and after disinfection (3S)). (**a**) Total bromine concentration versus TOC concentration. (**b**) Differences in absorbance versus differences in total bromine concentration.

Water quality tests performed in the water treatment plant and on the customers' premises by the State Sanitary Inspectorate revealed bromine contents well below the detectability levels. Good water quality after the modification of the treatment line is also evidenced by low THM concentrations (Figure 4).

Laboratory tests have revealed that the use of chlorine and ozone instead of sodium hypochlorite allows the amounts of thus formed THMs to be reduced by 98% [29]. The results of tests conducted at several WTPs indicate that the introduction of the pre-ozonation step results in a reduction in the total amounts of THMs in waters supplied to the public drinking water system [12,31–33].

The costs of construction of a mobile system for flushing and disinfection with ozone are several times lower than the costs of a standard ozonation plant. A standard ozone system installed at the outlet of the water to the tank would require ozonation contact chambers where the gas would be mixed with water and the mixing (reaction) time would be approximately 5 min. When using the mobile system, the highly ozonated water solution does not require the installation of contact tanks, and the mixing and reaction time of highly ozonated water with the product water can be shortened to 2 min due to a better and more effective water ozonation process. It is easier to precisely adjust the residual ozone to the value of 0.05 mg /L required by the law.

The benefits of using a mobile system for flushing and disinfection with ozone include:

- Less space needed for the assembly of the system;
- The possibility of moving to another place in the event of the necessity to use it on another section of the water supply network;
- Short investment time;
- Lower investment costs;
- Ease of system operation, process management and control;
- The ability to precisely dose ozone to water.

## 4. Conclusions and Future Perspectives

This study outlines the ozonation procedure introduced alongside the chlorination process using a small-size automatic ozonation installation. The results of water quality testing obtained under two different disinfection regimes were compared. The test results show that the introduction of the ozonation step alongside chlorination may prove to be a more effective method of disinfection and will not prompt the formation of DBPs at

concentrations above those of health concern. One has to bear in mind, however, that the group of hazardous by-products for which the formation potential is thus reduced is restricted here to THMs and bromates, the impacts of which have been well recognized and can be effectively controlled. Bromate occurrence identified in the ozonation processes will not cause health concerns as long as the regimes of disinfection with ozone and chlorine are maintained, even though the source waters may contain bromides at low concentrations.

Disinfection with sodium hypochlorite may result in the formation of increased amounts of bromine (in relation to its contents in raw water), including the occurrence of bromates in treated water to be supplied to the customers.

It was demonstrated that disinfection with sodium hypochlorite may result in the admission of bromine from those compounds that are not routinely determined in standard testing of the quality of water supplied to distribution systems.

In terms of water quality monitoring, it seems recommendable that concentrations of bromate compounds and by-products of the disinfection process should also be routinely determined in water treatment systems which do not involve ozonation.

In further research on water after disinfection, it would be appropriate to determine the concentrations of other disinfection by-products, with particular emphasis on bromine compounds.

To assess the effect of sodium hypochlorite on the formation of bromine disinfection by-products, it would be appropriate to determine the concentration of bromine in sodium hypochlorite, which is used in disinfection, as well as to determine the concentration of organic and inorganic forms of bromine in raw water. Research should be conducted taking into account seasonal variability.

The ozonation of water fed to the public water distribution system with the use of a small-sized fully automatic ozonation installation will not unduly complicate the water treatment process as there is no need to redesign the treatment line. Because of the relatively low investment costs involved, this technology can be well applied even in small WTPs. The environmental friendliness, high reliability and efficiency of the modified disinfection scheme ensure the high standards of water quality.

**Supplementary Materials:** The following supporting information can be downloaded at: https://www.mdpi.com/article/10.3390/w14182924/s1, Figure S1: Microbiology of surface waters (Skawinka River), test data provided by the State Sanitary Inspectorate (cfu—colony forming unit); Figure S2. Free chlorine in drinking water, supplied by the WTP Skawina, according to the report by State Sanitary Inspectorate title; Table S1: Microbiology of water supplied to the public water distribution network (data provided by the WTP Skawina 2017–2021; Table S2. Bromate concentrations in sodium hypochlorite and in treated waters.

**Author Contributions:** Conceptualization, B.W., R.M. and J.W.; methodology, B.W. and R.M.; validation, B.W., R.M. and J.W.; formal analysis, B.W.; investigation, B.W., R.M. and J.W.; resources, B.W., R.M. and J.W.; data curation, B.W., R.M. and J.W.; writing—original draft preparation, B.W.; writing—review and editing, B.W. All authors have read and agreed to the published version of the manuscript.

**Funding:** This research was funded by the AGH University of Science and Technology in Krakow [research subvention number 16.16.190.779].

**Institutional Review Board Statement:** Not applicable.

**Informed Consent Statement:** Not applicable.

**Data Availability Statement:** All data presented in this study are contained within the article.

**Conflicts of Interest:** The authors declare no conflict of interest.

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
