# Peer review of "Modification of Disinfection Process at a Local Water Treatment Plant—Skawina (Poland)"

_water, doi:10.3390/w14182924_

Round 1
Reviewer 1 Report (New Reviewer)
the topic is very interesting
a cost benefit analysis of the additional disinfection process is absolutely necessary
the findings of this analysis should be included in the results section and dicussed in the conclusions section
more comments can be found directly in the annotated manuscript

Author Response
Authors’ response: We want to thank for insightful review, which helped us correct and modify the paper. All comments and corrections of Reviewer #1 were included in the article. The list of modifications can be found below.
a cost benefit analysis of the additional disinfection process is absolutely necessary the findings of this analysis should be included in the results section and dicussed in the conclusions section.
Autors' response:
The cost benefit analysis could be the subject of a separate article. The main subject of our article was water quality after the modification of disinfection section, not economic analysis.
In accordance with the reviewer's remark, the text has been supplemented. At the end of the "Results and Discussion" chapter, some economic aspects of the mobile system and standard ozonation were compared.
more comments can be found directly in the annotated manuscript
Autors' response:
Page 1 line 35 ( the old version). The literature citation has been added.
Page 1 line 44 ( the old version). The literature citation has been added.
Page 3 line 101( the old version). The literature citation has been added.
Page 3 line 107 ( the old version). The text has been supplemented and citation has been added.
Page 9 line 360 (the old version). "Table S1" is correct
Page 11. In the place of "Admissable Value" should be “Parametric Value”
Page 12. Line 386. In accordance with the reviewer's remark, the text has been corrected.

Reviewer 2 Report (Previous Reviewer 3)
Greetings, Editor thank you for providing me with the opportunity to review the article. I reviewed the article with title Modification of Disinfection Process at the Local Water Treatment Plant - Skawina (Poland). The article topic is intriguing and promising in the area. Overall, the article structure and content are suitable for the Water journal. I am pleased to send you major level comments, there are some serious flaws which need to be corrected before publication. Please consider these suggestions as listed below.
1. The title seems good, but the abstract seems to be wired. Please add one more introductory line of your objective in beginning of abstract.
2. Research gap should be delivered on more clear way with directed necessity for the future research work.
3. Introduction section must be written on more quality way, i.e., more up-to-date references addressed.
4. The novelty of the work must be clearly addressed and discussed, compare previous research with existing research findings and highlight novelty.
5. What is the main challenge? Please highlight in the introduction part.
6. Please check the abbreviations of words throughout the article. All should be consistent.
7. Do not use lumpy reference please remove 1-4 and simple cite this one article here- Role of nanomaterials in the treatment of wastewater: a review.
8. Page 3 Line 107 need a strong reference, cite this article- Recent advances in metal decorated nanomaterials and their various biological applications: a review.
9. The main objective of the work must be written on the more clear and more concise way at the end of introduction section.
10. Please include all chemical/instrumentation brand name and other important specification.
11. Please add chemical reagents section and stated all chemical with brand specifications.
12. Please provide space between number and units. Please revise your paper accordingly since some issue occurs on several spots in the paper.
13. Results and discussion part seems ok
14. Regarding the replications, authors confirmed that replications of experiment were carried out. However, these results are not shown in the manuscript, how many replicated were carried out by experiment? Results seem to be related to a unique experiment. Please, clarify whether the results of this document are from a single experiment or from an average resulting from replications. If replicated were carried out, the use of average data is required as well as the standard deviation in the results and figures shown throughout the manuscript. In case of showing only one replicate explain why only one is shown and include the standard deviations.
15. Please provide high quality image for figure 6.
16. Please add a comparative profile section to compare your results than previous.
17. Conclusion section is missing some perspective related to the future research work, quantify main research findings, highlight relevance of the work with respect to the field aspect.
18. To avoid grammar and linguistic mistakes, Major level English language should be thoroughly checked. Please revise your paper accordingly since several language issue occurs on several spots in the paper.
19. Reference formatting need carefully revision. All must be consistent in one formate. Please follow the journal guidelines.
Author Response
Response to reviewers
Authors appreciate for all the effort and time put in the reviews.
Thank you very much for taking time to give us constructive suggestions, which are very helpful for us to improve our paper. We hope that our response will satisfy reviewers’ requests.
- The title seems good, but the abstract seems to be wired. Please add one more introductory line of your objective in beginning of abstract.
Authors’ response The first sentences of introduction are:
"This paper summarizes the studies undertaken at the water treatment plant in Skawina (WTP Skawina) where the disinfection process was modified by introducing a mobile ozonation system.......
The aim of thise study was to analyze whether the change of the disinfection method should affect the final water quality. The investigated water samples were treated in the mobile ozonation installation and in the disinfection process using sodium hypochlorite only”.
These sentences explained our objective. Because of the word limit, (200 words) it cannot be explained in more detail.
- Research gap should be delivered on more clear way with directed necessity for the future research work.
Authors’ response:
The last part of the introduction has been reorganized and supplemented. Research goals have been stated in the introduction (page 3, lines125-126, 130-138, 142-144)
The need for further research is mentioned in new sentences in last the part of the article "Conclusions and Future perspectives". (page 15, lines 541-548)
- Introduction section must be written on more quality way, i.e., more up-to-date references addressed.
Authors’ response: References have been improved and supplemented. The last part of the introduction has been reorganized.
- The novelty of the work must be clearly addressed and discussed, compare previous research with existing research findings and highlight novelty.
Authors’ response: According to the reviewer's remark, the novelty features contained in the article were described in more detail in the introduction (page 3, line `135-146).
To date no studies were undertaken to determine the effects of the altered disinfection system, the new treatment line incorporating the mobile ozonation installation enabling the simultaneous ozonation and chlorination. To the best of our knowledge there have been no publication on this subject. Therefore are studies cannot be compare with previous research. No measurements of total bromine and bromide concentrations were routinely taken at various points of the water treatment line in Poland. Consequently, results provided in publications on the subject, their concentrations were not listed alongside.
- What is the main challenge? Please highlight in the introduction part.
Authors’ response: Following the Reviewers’ suggestions, the introduction has been modified
The main challenge was written in these sentences:
“Here a strategy is proposed whereby the disinfection process modified through the use of a mobile installation” (page 3, line 125-126)
"The goal of study was to analyze whether the change of the disinfection method should affect the final water quality". (page 3, line 135-137)
"The ultimate goal of study was to establish whether the process is likely to prompt the formation of bromate and main bromine disinfection by-products" (page 3, line 142-144)
- Please check the abbreviations of words throughout the article. All should be consistent.
Authors’ response: Abbreviations have been checked, as suggested.
- Do not use lumpy reference please remove 1-4 and simple cite this one article here- Role of nanomaterials in the treatment of wastewater: a review.
Authors’ response: References have been improved and supplemented. The article: "Role of nanomaterials in the treatment of wastewater: a review" has been cited (page 2, line 33, page 2 line 39).
- Page 3 Line 107 need a strong reference, cite this article- Recent advances in metal decorated nanomaterials and their various biological applications: a review.
Authors’ response: The article : "Recent advances in metal decorated nanomaterials and their various biological applications: a review. ". has been cited (page 2, line 33.., page 2 line 39).
Page 3 line 107: The text: "Generally, the presence of bromates in water is mostly attributed to the disinfection with ozone even though ozonation may not be the decisive factor" is not relevant to the article: "Recent advances in metal decorated nanomaterials and their various biological applications: a review".
- The main objective of the work must be written on the more clear and more concise way at the end of introduction section.
Authors’ response: Introduction has been modified, as suggested (page 3, lines 123-146).
- Please include all chemical/instrumentation brand name and other important specification.
Authors’ response: At the moment it is not possible to include all chemical/instrumentation brand name and other important specification, moreover these do not seem to be of relevance. All instruments complied with the standards of the procedures listed in column 7 (Table 2).
- Please add chemical reagents section and stated all chemical with brand specifications.
Authors’ response: At the moment it is not possible to add the exact chemical reagents section and stated all chemical with brand specifications, moreover these do not seem to be of relevance. All chemical reagents complied with the standards of the procedures listed in column 7 (Table 2).
- Please provide space between number and units. Please revise your paper accordingly since some issue occurs on several spots in the paper.
Authors’ response: The manuscript has been revised, as suggested.
- Results and discussion part seems ok
- Regarding the replications, authors confirmed that replications of experiment were carried out. However, these results are not shown in the manuscript, how many replicated were carried out by experiment? Results seem to be related to a unique experiment. Please, clarify whether the results of this document are from a single experiment or from an average resulting from replications. If replicated were carried out, the use of average data is required as well as the standard deviation in the results and figures shown throughout the manuscript. In case of showing only one replicate explain why only one is shown and include the standard deviations.
Authors’ response: Water samples 1S, 2S, 3S i Ra). Were collected three times (on 8.03.2018, 05.08.2018, 06.05.2018). No statistical analysis has performed due to a limited number of samples.
- Please provide high quality image for figure 6.
Authors’ response:Figure 6 has been improved.
- Please add a comparative profile section to compare your results than previous.
To date no studies were undertaken to determine the effects of the altered disinfection system, the new treatment line incorporating the mobile ozonation installation enabling the simultaneous ozonation and chlorination. To the best of our knowledge there have been no publication on this subject. Similarly, no publications are available concerning the determination of total bromine and bromide concentration at various stages of the water treatment process. That is why the direct comparison of results is impossible.
- Conclusion section is missing some perspective related to the future research work, quantify main research findings, highlight relevance of the work with respect to the field aspect.
Autors' response: Conclusion section has been supplemented (page 15 lines 541-548).
- To avoid grammar and linguistic mistakes, Major level English language should be thoroughly checked. Please revise your paper accordingly since several language issue occurs on several spots in the paper.
Authors’ answer: The manuscript has been proof-read and revised, as suggested. We used editing service https://www.mdpi.com/authors/english. We have got the English editing certificate.
- Reference formatting need carefully revision. All must be consistent in one formate. Please follow the journal guidelines.
Authors’ response: Reference formatting has been revised.
- The title seems good, but the abstract seems to be wired. Please add one more introductory line of your objective in beginning of abstract.
Authors’ response The first sentences of introduction are:
"This paper summarizes the studies undertaken at the water treatment plant in Skawina (WTP Skawina) where the disinfection process was modified by introducing a mobile ozonation system.......
The aim of thise study was to analyze whether the change of the disinfection method should affect the final water quality. The investigated water samples were treated in the mobile ozonation installation and in the disinfection process using sodium hypochlorite only”.
These sentences explained our objective. Because of the word limit, (200 words) it cannot be explained in more detail.
- Research gap should be delivered on more clear way with directed necessity for the future research work.
Authors’ response:
The last part of the introduction has been reorganized and supplemented. Research goals have been stated in the introduction (page 3, lines125-126, 130-138, 142-144)
The need for further research is mentioned in new sentences in last the part of the article "Conclusions and Future perspectives". (page 15, lines 541-548)
- Introduction section must be written on more quality way, i.e., more up-to-date references addressed.
Authors’ response: References have been improved and supplemented. The last part of the introduction has been reorganized.
- The novelty of the work must be clearly addressed and discussed, compare previous research with existing research findings and highlight novelty.
Authors’ response: According to the reviewer's remark, the novelty features contained in the article were described in more detail in the introduction (page 3, line `135-146).
To date no studies were undertaken to determine the effects of the altered disinfection system, the new treatment line incorporating the mobile ozonation installation enabling the simultaneous ozonation and chlorination. To the best of our knowledge there have been no publication on this subject. Therefore are studies cannot be compare with previous research. No measurements of total bromine and bromide concentrations were routinely taken at various points of the water treatment line in Poland. Consequently, results provided in publications on the subject, their concentrations were not listed alongside.
- What is the main challenge? Please highlight in the introduction part.
Authors’ response: Following the Reviewers’ suggestions, the introduction has been modified
The main challenge was written in these sentences:
“Here a strategy is proposed whereby the disinfection process modified through the use of a mobile installation” (page 3, line 125-126)
"The goal of study was to analyze whether the change of the disinfection method should affect the final water quality". (page 3, line 135-137)
"The ultimate goal of study was to establish whether the process is likely to prompt the formation of bromate and main bromine disinfection by-products" (page 3, line 142-144)
- Please check the abbreviations of words throughout the article. All should be consistent.
Authors’ response: Abbreviations have been checked, as suggested.
- Do not use lumpy reference please remove 1-4 and simple cite this one article here- Role of nanomaterials in the treatment of wastewater: a review.
Authors’ response: References have been improved and supplemented. The article: "Role of nanomaterials in the treatment of wastewater: a review" has been cited (page 2, line 33, page 2 line 39).
- Page 3 Line 107 need a strong reference, cite this article- Recent advances in metal decorated nanomaterials and their various biological applications: a review.
Authors’ response: The article : "Recent advances in metal decorated nanomaterials and their various biological applications: a review. ". has been cited (page 2, line 33.., page 2 line 39).
Page 3 line 107: The text: "Generally, the presence of bromates in water is mostly attributed to the disinfection with ozone even though ozonation may not be the decisive factor" is not relevant to the article: "Recent advances in metal decorated nanomaterials and their various biological applications: a review".
- The main objective of the work must be written on the more clear and more concise way at the end of introduction section.
Authors’ response: Introduction has been modified, as suggested (page 3, lines 123-146).
- Please include all chemical/instrumentation brand name and other important specification.
Authors’ response: At the moment it is not possible to include all chemical/instrumentation brand name and other important specification, moreover these do not seem to be of relevance. All instruments complied with the standards of the procedures listed in column 7 (Table 2).
- Please add chemical reagents section and stated all chemical with brand specifications.
Authors’ response: At the moment it is not possible to add the exact chemical reagents section and stated all chemical with brand specifications, moreover these do not seem to be of relevance. All chemical reagents complied with the standards of the procedures listed in column 7 (Table 2).
- Please provide space between number and units. Please revise your paper accordingly since some issue occurs on several spots in the paper.
Authors’ response: The manuscript has been revised, as suggested.
- Results and discussion part seems ok
- Regarding the replications, authors confirmed that replications of experiment were carried out. However, these results are not shown in the manuscript, how many replicated were carried out by experiment? Results seem to be related to a unique experiment. Please, clarify whether the results of this document are from a single experiment or from an average resulting from replications. If replicated were carried out, the use of average data is required as well as the standard deviation in the results and figures shown throughout the manuscript. In case of showing only one replicate explain why only one is shown and include the standard deviations.
Authors’ response: Water samples 1S, 2S, 3S i Ra). Were collected three times (on 8.03.2018, 05.08.2018, 06.05.2018). No statistical analysis has performed due to a limited number of samples.
- Please provide high quality image for figure 6.
Authors’ response:Figure 6 has been improved.
- Please add a comparative profile section to compare your results than previous.
To date no studies were undertaken to determine the effects of the altered disinfection system, the new treatment line incorporating the mobile ozonation installation enabling the simultaneous ozonation and chlorination. To the best of our knowledge there have been no publication on this subject. Similarly, no publications are available concerning the determination of total bromine and bromide concentration at various stages of the water treatment process. That is why the direct comparison of results is impossible.
- Conclusion section is missing some perspective related to the future research work, quantify main research findings, highlight relevance of the work with respect to the field aspect.
Autors' response: Conclusion section has been supplemented (page 15 lines 541-548).
- To avoid grammar and linguistic mistakes, Major level English language should be thoroughly checked. Please revise your paper accordingly since several language issue occurs on several spots in the paper.
Authors’ answer: The manuscript has been proof-read and revised, as suggested. We used editing service https://www.mdpi.com/authors/english. We have got the English editing certificate.
- Reference formatting need carefully revision. All must be consistent in one formate. Please follow the journal guidelines.
Authors’ response: Reference formatting has been revised.

Round 2
Reviewer 1 Report (New Reviewer)
the paper has been adequately improved
BUT...the references section should be thoroughly crosschecked
Several corrections have been provided, but the authors are strongly recommended to carrefully check all references
especially those from Ochrona Åšrodowiska (7 in total) cannot be crosschecked. I suggest they all should be replaced by valid ones
Additionally the reference No#61 is not published yet

Author Response
Review 1
the paper has been adequately improved
BUT...the references section should be thoroughly crosschecked
Several corrections have been provided, but the authors are strongly recommended to carefully check all references
especially those from Ochrona Åšrodowiska (7 in total) cannot be crosschecked. I suggest they all should be replaced by valid ones
Additionally the reference No#61 is not published yet
Authors’ response:
We want to thank for insightful review, which helped improving the paper.
In accordance with the reviewer's remark, the part "References" has been corrected
Below you can find the list of corrections.
We added the issue numbers after the volumes numbers (references : No. 1 - line 577, No.5- line 587, No.6- line 589, No. 21- line 625, No.34- line 652, No. 35- line 655, No.38- line 662, No. 44- line 674, No. 45 - line 677, No. 66- lin2721.
But:
According to: https://www.mdpi.com/journal/water/instructions
References should be described as follows:
Journal Articles:
1. Author 1, A.B.; Author 2, C.D. Title of the article. Abbreviated Journal Name Year, Volume, page range.
- Links to the references: 8, 25, 27,32,42,51,62 have been added. It is easy to find them now.
- Mistakes have been corrected (6- line 588; 9- line594; 16- line 611)
- No. 61 There is the database . This database will not be published.

Reviewer 2 Report (Previous Reviewer 3)
Accepted
Author Response
Authors' response:
Authors are thankful for the reviewer's valuable comments and suggestions required for the improvement of the manuscript “Modification of Disinfection Process at a Local Water Treatment Plant —Skawina (Poland) "submitted for "Water".
Authors appreciate for all the effort and time put in the reviews. We are grateful for the acceptance of the manuscript.
With best regards,
Bogumiła Winid

This manuscript is a resubmission of an earlier submission. The following is a list of the peer review reports and author responses from that submission.
Round 1
Reviewer 1 Report
Modification of Disinfection Process at the Local Water Treatment Plant-Investigation of Water Quality in Skawina (Poland)
Dear authors,
This is an interesting study; however, there are several concerns which I would like to clarify before I suggest a decision on your manuscript.
I checked the similarity for the manuscript and found out it is less than 20%. Therefore, the paper can be moved to review. In addition, please pay attention to your language usage. There are some language slips and styling issues.
Title - "Modification of Disinfection Process at the Local Water Treatment Plant-Investigation of Water Quality in Skawina (Poland)" - Please rephrase it to make a simple title. The language in it at present is not clear.
Abstract - Where is your research gap in the abstract? I think, it is missing here. Why do you want to do this research? Ozoning is a quite common method. So what is your input to the research work? This is also not clear from your abstract.
Introduction - Good number of references are cited here. However, please check if the referencing whether it is correctly sequenced or not.
"The aim of this paper is to explore the potentials of water disinfection with ozone as a strategy to minimise the chlorine doses, recalling the case study of the water treatment plant in Skawina (WTP Skawina)."
What is the new of this? Is it the application to the specific treatment plant?
Materials and Methods - Can the authors include a location map in this section since this manuscript is case study based?
Pay attention to units. 352.4 km2 has to be 352.4 km2
KMnO4 - KMnO4
Figure 1 is unreadable. Please make it clear.
"Water samples were collected in accordance with the obligatory procedure and physico-chemical parameters of water were determined by the recommended reference methodology [43,44]." - How many samples? What are the statistic of these samples? How did you test them? Are these based on single sample?
2.3. Ozonation step as a modification to the water disinfection process - what is your research input? Very unclear other than a project!
Results and discussion - You have presented a lot of results in this table. Selected physico-chemical parameters of water from the WTP Skawina treated with ozone 341 (step 3- sample collected on 06/08/2018). However, your research input is very doubtful.
Can authors presented their research input?
Conclusions - Is this the first time to use O3 into water treatment? No sound conclusions are made.
Reviewer 2 Report
1. Content of Introduction is short. Research background is missing.
2. What is the new contribution in your work
3. State research gap & explain reason for doing this research work
4. What is the ultimate goal of your work
5. Give citation wherever required
6. What is the limitations of your study?
7. Describe the innovation and goals of this research to differentiate your work from others.
8. Methodology or process is not clearly explained.
9. Conclusion & abstract is vague.
10. Add some below mentioned papers:-
-- Chaudhari, A. N., Mehta, D. J., & Sharma, N. D. (2022). Coupled effect of seawater intrusion on groundwater quality: study of South-West zone of Surat city. Water Supply, 22(2), 1716-1734.
-- Sharma, R. K., Khan, N., & Shukla, S. K. (2022). Evaluation of Groundwater Quality and its Suitability Assessment for Drinking and Agriculture purposes in Vidarbha Region of Maharashtra, India. Journal of Indian Association for Environmental Management (JIAEM), 42(1), 1-10.
Reviewer 3 Report
Greetings, Editor thank you for providing me with the opportunity to review the article. I reviewed the article with title Modification of Disinfection Process at the Local Water Treatment Plant-Investigation of Water Quality in Skawina (Poland). The article topic is intriguing and promising in the area. Overall, the article structure and content are suitable for the WATER journal. I am pleased to send you major level comments, there are some serious flaws which need to be corrected before publication. Please consider these suggestions as listed below.
1. The title seems good, but the abstract seems to be wired. Please add one more introductory line of your objective in beginning of abstract.
2. Introduction should be in one heading please merge section 1.1. into main section.
3. Research gap should be delivered on more clear way with directed necessity for the future research work.
4. Introduction section must be written on more quality way, i.e., more up-to-date references addressed. Please target the specific gap such as 2015-2021 etc.
5. The novelty of the work must be clearly addressed and discussed, compare previous research with existing research findings and highlight novelty.
6. What is the main challenge? Please highlight in the introduction part.
7. Please check the abbreviations of words throughout the article. All should be consistent.
8. Please include all chemical/instrumentation brand name and other important specification.
9. Page 1, Line 34 need to cite this article- Role of nanomaterials in the treatment of wastewater: a review.
10. Page 1, Line 40, please cite this article (Recent advances in metal decorated nanomaterials and their various biological applications: a review).
11. The main objective of the work must be written on the more clear and more concise way at the end of introduction section.
12. Please provide space between number and units. Please revise your paper accordingly since some issue occurs on several spots in the paper.
13. Regarding the replications, authors confirmed that replications of experiment were carried out. However, these results are not shown in the manuscript, how many replicated were carried out by experiment? Results seem to be related to a unique experiment. Please, clarify whether the results of this document are from a single experiment or from an average resulting from replications. If replicated were carried out, the use of average data is required as well as the standard deviation in the results and figures shown throughout the manuscript. In case of showing only one replicate explain why only one is shown and include the standard deviations.
14. Please provide high quality image for figure 1,3, and 5.
15. Please add a comparative profile section to compare your results and prove how it better than previous.
16. Section 4 should be renamed by Conclusion and Future perspectives. Conclusion section is missing some perspective related to the future research work, quantify main research findings, highlight relevance of the work with respect to the field aspect. In present form chapter 4 is totally weird.
17. To avoid grammar and linguistic mistakes, Major level English language should be thoroughly checked. Please revise your paper accordingly since several language issue occurs on several spots in the paper.
18. Reference formatting need carefully revision. All must be consistent in one formate. Please follow the journal guidelines.